# The Diagnostic Accuracy of Magnetic Resonance Imaging for Maternal Acute Adnexal Torsion during Pregnancy: Single-Institution Clinical Performance Review

**DOI:** 10.3390/jcm9072209

**Published:** 2020-07-13

**Authors:** Jong Hwa Lee, Hyun Jin Roh, Jun Woo Ahn, Jeong Sook Kim, Jin Young Choi, Soo-Jeong Lee, Sang Hun Lee

**Affiliations:** 1Department of Radiology, University of Ulsan College of Medicine, Ulsan University Hospital, 44033 Ulsan, Korea; jhlee@uuh.ulsan.kr; 2Department of Obstetrics and Gynecology, University of Ulsan College of Medicine, Ulsan University Hospital, 44033 Ulsan, Korea; 0729345@uuh.ulsan.kr (H.J.R.); ahnjwoo@uuh.ulsan.kr (J.W.A.); jeongsookkim@uuh.ulsan.kr (J.S.K.); 0734208@uuh.ulsan.kr (J.Y.C.); exsjlee@uuh.ulsan.kr (S.-J.L.)

**Keywords:** diagnostic accuracy, magnetic resonance imaging, acute adnexal torsion, pregnant women, ADC values, corpus luteal cystic ovary, cystectomy detorsion, salpingo-oophorectomy

## Abstract

Background: For acute adnexal torsion of pregnant women, appropriate treatment based on an accurate diagnosis is especially important for fertility preservation and timely treatment. The 2017 American College of Obstetricians and Gynecologists (ACOG) Committee Opinion No. 723 announced its practice-changing guidelines to ensure that diagnostic magnetic resonance imaging (MRI) conducted during the first trimester and gadolinium exposure at any time during pregnancy are safe for fetal stability. Unfortunately, few studies have been performed to evaluate the usefulness of the diagnostic accuracy of MRI for acute adnexal torsion during pregnancy. Objective: We sought to determine the efficacy of diagnostic MRI modality using multiparameter for maternal adnexal torsion during pregnancy. Methods: From 1 January 2007 to 31 January 2019, 131 pregnant with MRI tests were reviewed. In this retrospective cohort study, 94 women were excluded due to conditions other than an adnexal mass, and 37 were identified through MRI analyses conducted before surgery for suspected adnexal torsion. The primary outcome was the diagnostic accuracy of sonography and MRI, and the secondary outcome was the usefulness of Apparent diffusion coefficient (ADC) values for predicting the severity of hemorrhagic infarction between the medulla and cortex of the torsed ovarian parenchyma. Results: Our study demonstrates that in the diagnosis of adnexal torsion during pregnancy, the sensitivity, specificity, positive predictive value, and negative predictive value are 62.5%, 83.3%, 90.9%, and 45.5% for sonography and 100%, 77.8%, 90.5%, and 100% for MRI. MRI results in surgical-proven adnexal torsion patients revealed unilocular ovarian cysts (36.8% (7/19)), multilocular ovarian cysts (31.6% (6/19)), and near normal-appearing ovaries (31.6% (6/19)). Pathology in adnexal torsion revealed a corpus luteal ovarian cyst (63.2% (12/19)) and underlying adnexal pathology (46.8% (7/19)). Maternal adnexal torsion during pregnancy was more likely to occur in corpus luteal ovarian cysts than in underlying adnexal masses (odds ratio, 2.14; 95% confidence interval (CI), 0.428–10.738). MRI features for adnexal torsion were as follows: tubal wall thickness, 100% (19/19); ovarian stromal (medullary) edema, 100% (19/19); symmetrical or asymmetrical ovarian cystic wall, 100%(19/19); prominent follicles in the ovarian parenchyma periphery, 57.9% (11/19); periadenxal fat stranding, 84.2% (16/19); uterine deviation to the twisted side, 21.1% (4/19); and peritoneal fluid, 42.1% (8/19). The signal intensity of the ADC values of the ovarian medulla and cortex were compared between the cystectomy and detorsion (CD) and salpingo-oophorectomy (SO) groups. The ADC values of the CD and SO groups were 1.81 ± 0.09 × 10^−3^ mm^2^/s and 1.91 ± 0.18 × 10^−3^ mm^2^/s, respectively (*P* = 0.209), in the ovarian medulla and 1.37 ± 0.32 × 10^−3^ mm^2^/s and 0.96 ± 0.36 × 10^−3^ mm^2^/s, respectively (*P* = 0.022), in the ovarian cortex. The optimal cut-off value of ADC values for predictable total necrosis in the torsed ovarian cortex was ≤ 1.31 × 10^−3^ mm^2^/s (area under the curve (AUC) = 0.81; 95% CI 0.611–1.0; *P* = 0.028). Conclusion: Our data showed that maternal adnexal torsion during pregnancy occurred in most corpus luteal cystic ovary cases and some normal-appearing ovary during the 1st and 2nd trimesters of gestation. Therefore, this study is the first study to elaborate on the existence or usefulness of the diagnostic MRI for acute maternal adnexal torsion during pregnancy and to provide a predictive diagnosis of the severity of hemorrhagic infarction for deciding surgical radicality.

## 1. Introduction

Maternal adnexal torsion in pregnancy is considered relatively rare, with an incidence of 1–5: 10,000 in natural pregnancy. However, the incidence has soared to 7.5–16% in cases of ovarian hyperstimulation related to associated reproductive technology (ART) [1,2].

In contrast to that in nonpregnancy, the prevalence of adnexal torsion in pregnancy is related to gestational age, and the disease occurs more frequently in the first and early second trimesters than in the third trimester. The most common ovarian etiology in these periods reveals a corpus luteal cyst that is related to normal-appearing ovaries in as many as 46% of pediatric and adolescent cases, unlike the usual adult cases in which occur more commonly in the context of a pre-existing adnexal mass [3,4,5,6,7].

An accurate diagnosis is particularly important for pregnant women with adnexal torsion because treatment decisions based on the diagnosis are associated with a risk of damage to the reproductive organs by necrosis, early missed abortions, and life-threatening fetal events due to preterm labor if surgical treatment is not performed at the optimal time [5,6,7,8,9]. Therefore, obtaining an accurate diagnosis of maternal adnexal torsion during pregnancy is one of the challenges faced by obstetricians and gynecologists.

Some authors argue the superiority of first-line magnetic resonance imaging (MRI) to ultrasonography (USG) in nonpregnant women with adnexal torsion. However, diagnostic tools including USG, [10] Doppler flow imaging, [11,12,13] computerized tomography (CT), and MRI [14,15,16,17] still lack reliable diagnostic significance to confirm preoperative adnexal torsion, regardless of pregnancy status. Regarding suspected adnexal torsion in adolescents, Practice guidelines of the American College of Obstetricians and Gynecologists (ACOG) Committee Opinion recommend that the definitive diagnosis is surgical confirmation through laparoscopic exploration [18,19]. For strongly suspected adnexal torsion in pregnancy, the approach to timely intervention with diagnostic laparoscopy by a skillful surgeon may be similar to that used when treating nonpregnant women.

However, because adnexal torsion in pregnancy is a particularly unique disease entity, the diagnostic modality of USG, as well as MRI, for adnexal torsion in pregnancy has been assessed mainly in anecdotal case reports and small case series [15,16]. Moreover, there seems to be no study that has attempted to evaluate the accuracy of USG or MRI in maternal adnexal torsion during pregnancy.

This study shows that the goal of MRI is to obtain a precise diagnosis that guides the therapeutic decision regarding the most appropriate treatment.

## 2. Materials and Methods

### 2.1. Study Protocol

A total of 131 pregnant women who underwent MRI due to abdominal pain in the ER, obstetrics, and gynecology department or other departments, from 1 January 2007 to 31 January 2019, were examined in this study. The patient was included if the urine pregnancy test was positive and if sonography at the time of the MRI showed a viable fetus. Women were excluded if they were not pregnant or if they were examined by MRI for reasons other than suspected adnexal torsion. Ninety-four patients were excluded due to reasons other than adnexal mass. Thirty-seven pregnant women identified through MRI analyses conducted before the surgery for suspected adnexal torsion were identified (Figure 1). This study was approved by the ulsan university hospital institutional review board (IRB UUH 20-06-039).

### 2.2. MRI

#### 2.2.1. MRI Protocol

All MRI examinations were carried out on a 3T MRI scanner without the use of intravenous gadolinium-based contrast agents. (Intera Achieva 3T, Philips Medical System, Best, The Netherlands). Multiparametric imaging sequence parameters included multiplanar SPIR (spectral presaturation with inversion recovery) fat-suppressed T1-weighted (T1W) imaging, T2-weighted (T2W) single-shot turbo spin-echo (SSH-TSE) imaging, SPAIR (spectral attenuated inversion recovery) fat-suppressed T2W SSH-TSE imaging, and diffusion-weighted imaging (DWI). SPIR T1W MRI only in the axial plane, T2W SSH-TSE imaging in three planes, and SPAIR T2-weighted SSH-TSE imaging in the axial and coronal planes were performed in all patients. DWI was performed using a single-shot spin-echo echo-plana imaging sequence with a chemical shift-selected pulse sequence (TR/TE 2200-4000/90-120). Images were acquired using a motion-proving gradient pulse of the same strength applied sequentially along three orthogonal directions) with the use of two b values (0 and 1000 s/min^2^). Apparent diffusion coefficient (ADC) values were determined using the two b values, as mentioned earlier.

#### 2.2.2. MRI Interpretation

A retrospective analysis of electronic clinical and MRI databases by picture archiving and communication system (PACS) verified by clinicians and radiologists involved in this study was conducted at our department (University of Ulsan College of Medicine, Ulsan University Hospital, Ulsan, Korea). Our hospital has 24 h of MRI scanner availability and is divided into regulation time (9 AM–5 PM) and nonregulation time (5 PM–9 AM), which is covered by the emergency room. All MR images from regulation time hours were reviewed by an attending radiologist (J.Y.L.) with experience in the abdominal and gynecologic field, and an on-call radiologist generated preliminary reports during nonregulation time. The radiologist interpreted the images based on the presence of the following MRI features (MRI features are listed in Appendix A).

In addition, for quantitative analysis, the ADC values were measured in the ovarian stroma (medulla) using regions of interest (ROIs) of round or oval shape and approximately 30 mm^2^ in size, excluding cystic or follicular components as much as possible. ADCs were also measured in the cortex, containing multiple peripheral follicles, using ROIs placed in the lesion grossly revealing the lowest ADC value, while excluding the central cystic zone of prominent follicles and cystic or medullary components as much as possible. Each ADC was determined as the mean value of ROI measurements of three lesion foci.

#### 2.2.3. USG

Preoperative US images were examined at two different places in our department (gynecologic emergency room, obstetrics, and gynecologic US unit) using different ultrasound machines (Voluson Expert, Voluson E8, S8, and S10, all by GE Healthcare, Milwaukee, WI, USA). All US machines were performed using a transvaginal probe (5–7.5 MHz, within a range of 6 cm from the transducer tip) and a transabdominal probe (3.5–5 MHz). Color Doppler flow parameters were optimized (spatial peak temporal average intensity, 40–92 MW/cm^2^; wall filter, 130–500 Hz; pulse repetition frequencies, between 1 and 11 kHz; velocity ranges lowered to 4 cm/s; and Doppler angle of insonation <60).

#### 2.2.4. USG Interpretation

All preliminary USG studies were performed by qualified sonographers and obstetric and gynecologic residents. If the torsed ovary and its twisted vascular pedicle appeared abnormal on both gray-scale and color Doppler imaging, the patient was also evaluated by a maternal-fetal medicine subspecialist with specific expertise in USG. Gray-scale sonographic findings that were assessed in this study had the following diagnostic criteria: (a) whirlpool sign (ring of pearls sign), (b) coexistent mass with the twisted ovary, (c) the presence of a twisted vascular pedicle, and (d) the presence of free pelvic fluid, a unilateral enlarged ovary (>4 cm) with hyperechoic stromal edema and peripherally displaced follicles.

### 2.3. Data Analysis

The data are presented frequency and percentage for categorical variables and mean ± standard deviation (SD) for numeric variables. Differences in study participants’ characteristics were compared across subgroups with χ2 test or Fisher’s exact test for categorical variables and independent test or Mann–Whitney’s U test for continuous variables as appropriate. Differences in study participants’ characteristics were compared across subgroups with the analysis of variance (ANOVA) with Scheffe’s post-hoc test or Kruskal–Wallis test with Dunn’s post-hoc test as appropriate. To check if its distribution is normal, we used Shapiro–Wilk’s test. The inter-rater reliability of the two raters’ classifications was assessed by kappa statistic. The receiver operating characteristic (ROC) curve was performed to assess the sensitivity and specificity of ADC values for predicting prediction of the severity of hemorrhagic infarction in the torsed ovary. For data visualization, the box plot was also displayed. All statistical analyses were carried out using SPSS 24.0, MedCalc 16.4.3, and R 3.6.2, and p values less than 0.05 were considered statistically significant.

## 3. Results

### 3.1. Participant Characteristics

The cohort was described, and baseline characteristics were compared for pregnant women diagnosed with adnexal/nonadnexal torsion by MRI. We confirmed adnexal torsion in 19 patients (51.4%), and nonadnexal torsion in 9 patients (24.3%) by diagnostic surgery; however, whether adnexal torsion/nonadnexal torsion was present could not be surgically determined in 9 (24.3%) patients. The characteristics of the participants in the three groups regarding adnexal torsion versus nonadnexal torsion and surgically unavailable adnexal/nonadnexal torsion data are shown in Table 1.

In the adnexal torsion group, radical treatment of five cases (adnexectomy including salpingectomy) was decided because the adnexa appeared to have a total necrotic appearance or did not return to a viable-looking appearance after detorsion of the ischemic adnexa. In eleven other cases, conservative treatment (cystectomy or detorsion) was applied because of a partial necrotic appearance (Figure 2). Two women who had rare cases of torsion in the salpinx underwent salpingectomy at 10.2- and 35.2-weeks gestational age (Figure 3). The final pathology-based diagnoses were corpus luteum cyst (*n* = 12), serous cystadenoma (*n* = 2), tubal cyst (*n* = 2), mucinous cystadenomas (*n* = 2), and mature cyst teratoma (*n* = 1).

This study revealed unilocular ovarian cysts in 36.8% (7/19), multilocular ovarian cysts in 31.6% (6/19), and near normal-appearing ovaries in 31.6% (6/19) on preoperative MRI. The pathology results revealed a corpus luteal ovarian cyst in 63.2% (12/19) and underlying adnexal pathology in 46.8% (7/19).

In the nonadnexal torsion group, radical treatment of one case (adnexectomy) was decided due to ovarian malignancy. The final pathology-based diagnoses were corpus luteum cyst (*n* = 4), mature cyst teratoma (*n* = 1), serous cystadenoma (*n* = 1), mucinous cystadenomas (*n* = 2), and serous adenocarcinoma (*n* = 1).

This study evaluated the obstetric outcomes of pregnancies after the management of the adnexal torsion. Three patients with adnexal torsion had preterm labor after the operation, which regressed in 4 ± 0.25 days (range 3–5) under tocolytic treatment. The adnexa showed a total necrotic appearance in one case of these patients, and there were partial ischemic lesions in 2 cases.

### 3.2. MRI and Ultrasonography

With the use of our imaging protocol, the sensitivity, specificity, PPV, and NPV were 62.5%, 83.3%, 90.9%, and 45.5%, respectively, for USG and 100%, 77.8%, 90.5%, and 100%, respectively, for MRI (Table 2).

All 37 women (100%) with an adnexal mass in this study underwent USG before MRI. The rate of a visualized ovary on USG was 75.7% (28/37). Of the 28 women with a visualized ovary, 5 women did not undergo Doppler flow imaging (Appendix A).

The rate of a nonvisualized ovary on USG was 24.3% (9/37). Of 9 women with nondiagnostic USG, 4 women (44%) were diagnosed with adnexal torsion on MRI (Appendix A).

The quantitative results regarding ADC values in the ovarian medulla and cortex of the cystectomy and detorsion (CD) and salpingo-oophorectomy (SO) groups, respectively, were estimated in Table 3. The ADC values of the CD and SO groups in the ovarian medulla were 1.81 ± 0.09 × 10^−3^ mm^2^/s and 1.91 ± 0.18 × 10^−3^ mm^2^/s, respectively (*P* = 0.209). The ADC values of the CD and SO groups in the ovarian cortex were 1.37 ± 0.32 × 10^−3^ mm^2^/s and 0.96 ± 0.36 × 10^−3^ mm^2^/s, respectively (*P* = 0.022). Box-and-whisker plot of ADC values for partial and total necrosis between ovarian medulla and cortex, respectively, confirmed significant differences in ovarian cortex groups (*P* = 0.022) (Figure 4).

Figure 5 shows the receiver operating curves of ADC values between ovarian medulla and cortex to predict the severity of hemorrhagic infarction in the torsed maternal ovary during pregnancy. The suitable threshold cut-off value of ADC values for total necrosis in torsed ovarian cortex is ≤1.31 × 10^−3^ mm^2^/s (area under the curve (AUC) = 0.81; 95% CI 0.611–1.0; *P* = 0.028).

Appendix A summarizes the interobserver variability for the MRI features. The results of the observers showed substantial to complete agreement regarding all of the imaging findings.

## 4. Discussion

### 4.1. Principal Findings

Our study found that maternal adnexal torsion during pregnancy occurred in most corpus luteal cystic ovary cases in the 1st and 2nd trimesters of gestation and, in some cases, near the normal-appearing ovary and that the accuracy of MRI diagnosis for adnexal torsion in pregnant women is high. In the clinical setting, enlarged ovaries with swollen medullary edema, prominent peripheral follicles with thickened walls, and restricted diffusion in the cortex were the main imaging findings of acute adnexal torsion with hemorrhagic infarction. In our study, ADC signal intensity was confirmed to predict the severity of hemorrhagic infarction in the torsed ovary through the differences in decreased signal intensity between the medulla and cortex of the torsed ovarian parenchyma (Figure 6). Thus, MRI modality is useful for diagnosing suspected acute adnexal torsion during pregnancy.

### 4.2. Fluctuation Situation of ACOG Committee Guidelines for Diagnostic Imaging During Pregnancy

Recently, the ACOG Committee has begun paying attention to the uncertainty surrounding the long-term assessment of fetal stability related to maternal diagnostic MRI according to gestational age in pregnancy. According to the ACOG guidelines before the 2017 announcement, although MRI during pregnancy did not have any fetal adverse effects, such as teratogenesis, carcinogenesis, or acoustic risk, the guidelines suggest that avoiding MRI during the first trimester is good practice [20,21].

However, the recent ACOG Committee Opinion No. 723, published in 2017, provides its practice-changing guidelines to ensure that diagnostic MRI conducted during the first trimester and gadolinium exposure at any time during pregnancy are safe for fetal stability, based on evidence recently published in JAMA [20,22].

### 4.3. Comparison with Other Studies

The pathophysiology of adnexal torsion in premenarchal females (children or premenarchal adolescents) who frequently experience adnexal torsion in the context of normal ligamentous laxity tends to differ from that of pregnant women who involve changes in ligamentous laxity due to uterine enlargement.

The combination of a corpus luteal cystic ovary secreting progesterone hormones during the first eight weeks of pregnancy, as well as an ovary with a preexisting tumor, and the rapid anatomical changes of the pelvis in the 1st and 2nd trimesters of gestation will contribute to an increased risk of adnexal torsion during pregnancy with underlying pathophysiology.

According to our results, all adnexal torsion rates during the combined period of the first (63.2%) and second trimesters (21.1%) account for 84.3% of the total, which is somewhat lower than the rates of 100% and 94.2% previously published studies by Hibbrnd et al. and Smorgick et al., respectively [6,14]. In this study, MRI results in the adnexal torsion group confirmed by surgery revealed unilocular ovarian cysts in 36.8% (7/19), multilocular ovarian cysts in 31.6% (6/19), and near normal-appearing ovaries in 31.6% (6/19).

In Pansky et al., it was defined as “torsion of normal adnexa” when enlargement, suspicious adnexal masses, or cysts were not demonstrated on preoperative imaging modalities or intraoperative examination [23]. However, there have been no previous reports evaluating the size of the “torsion of normal adnexa.” Some authors have shown that most cases of adnexal torsion are associated with unilateral enlarged ovaries of more than 4cm [24,25]. Therefore, in our series, the so-called “normal-appearing ovaries with or without infarction in adnexal torsion” were defined as less than 4-cm ovaries without suspicious adnexal masses or cysts [23]. Of 6 cases of near normal-appearing ovaries, five cases occurred in the first trimester at 8.3, 10.2, 10.6, 11.6, and 13.5 weeks of gestational age, with 1 case occurring in the third trimester at 35.2 weeks (ovarian sizes of 4, 3.5, 3.5, 3, 4, and 3.5 cm in ovarian size, respectively).

On sonography, but not MRI, Smorgick et al. showed unilocular ovarian cysts in 39.5% (15/38), multicystic ovaries in 36.8% (14/38), and near normal-appearing ovaries in 23.7% (9/38) cases. Whereas “normal” ovarian torsion was more common in the second and third trimesters than in the first trimester (6 cases vs. 3 cases, respectively). The reason for the differing results between the Smorgick et al. study mentioned above and our study is likely due to differences related to the population of OHSS patients who conceived by ovulation induction or in vitro fertilization (48.5% vs. 21.1%) [6].

Our pathology in adnexal torsion revealed a corpus luteal ovarian cyst in 63.2% (12/19) and underlying adnexal pathology in 46.8% (7/19). Maternal adnexal torsion during pregnancy was more likely to occur in corpus luteal ovarian cysts than in underlying adnexal masses (odds ratio, 2.14; 95% confidence interval, 0.428–10.738).

Adnexal torsion during pregnancy can occur in the ovary with corpus luteal cystic at the 1st and 2nd trimesters of gestation [7] and, in some cases, near the normal-appearing ovary (Figure 2).

### 4.4. Efficacy of Diagnostic MRI for Maternal Adnexal Torsion during Pregnancy

Our study demonstrates that sonography and MRI have sensitivity, specificity, positive predictive value, and negative predictive value in the diagnosis of adnexal torsion during pregnancy (62.5%, 83.3%, 90.9%, 45.5% vs. 100%, 77.8%, 90.5%, 100%, respectively).

One series examined the effectiveness of US scanning in diagnosing adnexal torsion, regardless of whether the patients were adolescents, and showed a PPV of 87.5% and specificity of 93.3% [26]. However, the US results reported in many studies now show highly variable misdiagnoses, [27,28,29,30,31] and Doppler flow imaging alone is not sufficient to confirm the preoperative diagnosis of adnexal torsion.

Gray-scale sonographic findings that assessed the diagnosis of adnexal torsion had the following diagnostic criteria: (a) whirlpool sign (ring of pearls sign), (b) coexistent mass with the twisted ovary, (c) the presence of a twisted vascular pedicle, and (d) the presence of free pelvic fluid, a unilateral enlarged ovary (>4 cm) with hyperechoic stromal edema and peripherally displaced follicles. Some of these ultrasound findings may exist as limited findings during pregnancy, such as the edematous appearance of the stroma in multicystic ovaries in women treated for ovulation induction. Ultrasound also has technical difficulties in assessing the ovaries of advanced pregnancies.

As the pregnancy progressed, the pregnant women’s ovaries shrank or were pushed out of the pelvis by the gravid uterus, making visualization more difficult. In our series, all 9 women with a nonvisualized ovary on USG showed MRI features with a visualized ovarian mass or adnexal mass in the 2nd and 3rd trimesters (Appendix A).

### 4.5. MRI Findings by a Multiparameter Imaging

The MRI features of specific adnexal torsion, such as the twisted pedicle and prominent peripheral follicles with the thickened wall, as well as nonspecific findings such as enlarged ovary with swollen medullary edema, are the same as those seen on CT.

Rha et al., reported that the MRI features of adnexal torsion in 25 patients showed for the following: fallopian tube thickening, 84% (21/25); smooth wall thickening of the twisted adnexal cystic mass, 82.6% (19/23); ascites 64% (16/25); and uterine deviation to the torsed side, 36% (9/25). The MRI features of this study are summarized in Table 3. Tubal wall thickness, ovarian stromal (medullary) edema, and symmetrical or asymmetrical ovarian cystic wall thickness were shown in all of our MR images.

However, the important advantage of MRI in the diagnosis of acute adnexal torsion is its demonstration of hemorrhagic infarction lesions by a better depiction of soft-tissue contrast. In the torsed ovary with subacute blood products, a high signal on T1W, T2W, and T1W fat-saturated sequences indicates hemorrhagic infarction. DWI can also be used as the imaging modality and best describes early ischemia [32,33]. Some authors have postulated that T2W contrast information of the torsed ovarian stroma may be useful in predicting the severity of hemorrhagic infarction [15,16,17].

In Kato et al.’s series, the ADC signal intensity was significantly lower in patients with ovarian torsion with hemorrhagic infarction than in those without infarction [33].

The gynecologic surgeon performed radical treatment (adnexectomy) when the adnexa looked necrotic or did not return to a viable-looking appearance after detorsion of the ischemic adnexa. Therefore, it is crucial to predict the radicality of the ovary in fertile women, before surgery.

In this study, ADC signal intensity confirmed to predict the severity of hemorrhagic infarction in the torsed ovary through the various differences in decreased signal intensity between the medulla and cortex of the torsed ovarian parenchyma.

We found that ADC signal intensity was not much different in the swollen medullary parenchyma of the conservative treatment group and radical treatment group. However, it was significantly lower in the cortical parenchyma with prominent follicles in the radical treatment group. The optimal cut-off value of ADC values for partial or total necrosis of the torsed ovarian cortex was first estimated using receiver operating characteristic curves.

### 4.6. Study Limitations

This study has several limitations. First, this study is retrospective, with a small sample size, and leads to associated selection biases. Second, this study was conducted in a single center. Although a study based on a single center obtains benefits in terms of internal control, there are interobserver biases.

Additionally, despite our relatively small sample size, the overall rarity of adnexal torsions in pregnant women limited our precision in estimating the diagnostic rates of MRI for detecting adnexal torsion in this context.

## 5. Conclusions

MRI has usually been used in subacute cases of ambiguous presentation among pregnant women with suspected adnexal torsion, and using MRI as the first-line modality without USG can be against the recommendation. Unlike those for adolescent women, treatment guidelines for adnexal torsion during pregnancy, ranging from the imaging diagnosis modality to surgical treatment, are still controversial due to the rarity of this condition in pregnancy. These data suggest that MRI is useful not only for identifying or excluding adnexal torsion during pregnancy but also for providing a predictive diagnosis to decide further treatment.

## Figures and Tables

**Figure 1 jcm-09-02209-f001:**
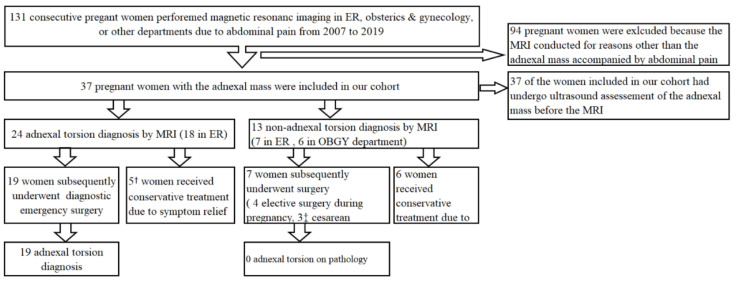
Flow diagram of the study participants. † One woman underwent ultrasound-guided aspiration of an ovarian cyst at 30.5 weeks’ gestation; two women did not undergo surgical extirpation of the mass, but the remaining two women (adnexal torsion MRI diagnosis at 5.2 and 9.5 weeks, respectively) confirmed no adnexal torsion during appendectomy with appendicitis (at 16.6 and 11.1 weeks, respectively) in pregnancy. ‡ A total of 3 women confirmed no adnexal torsion through simultaneous adnexal operation during cesarean section (1 emergent c/sec at 36.4 weeks, and 2 elective c/sec at 36.3 and 39.1 weeks, respectively). MRI, magnetic resonance imaging; ER, Emergent room; OBGY, obstetrics and gynecology.

**Figure 2 jcm-09-02209-f002:**
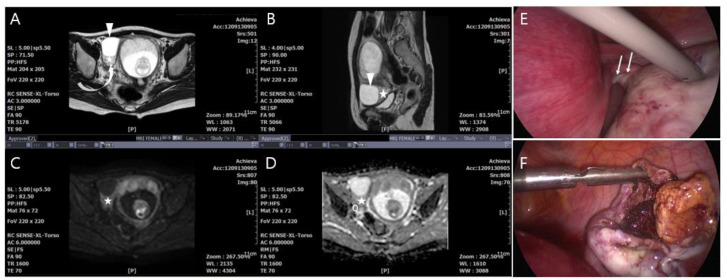
A 30-year-old pregnant woman (8 weeks) with acute adnexal torsion without hemorrhagic infarction in the cystectomy and detorsion group. (**A**) and (**B**) Transaxial T2-weighted single-shot turbo spin-echo MR images show hyperintense swollen ovarian medullary stroma (star) and prominent cortical follicle (curved arrow) in torsed enlarged right ovary; 3.5 cm sized unilocular cystic mass (arrowhead) in the normal-appearing ovary was pathologically confirmed as a corpus luteal cyst. (**C**) Transaxial diffusion-weighted MR image shows hyperintense swollen ovarian medullar (star). (**D**) Apparent diffuse coefficient (ADC) value indicating the region of interest (ROI) was 1.83 ± 0.11 × 10^−3^ mm^2^/s in the ovarian medulla (star) and 1.57 ± 0.14 × 10^−3^ mm^2^ in the ovarian cortex (circle), respectively. (**E**) Gross appearance of the ovarian pedicle with 720° torsion (arrows) under laparoscopy. (**F**) Cystectomy and detorsion was performed by surgical methods of laparoscopy.

**Figure 3 jcm-09-02209-f003:**
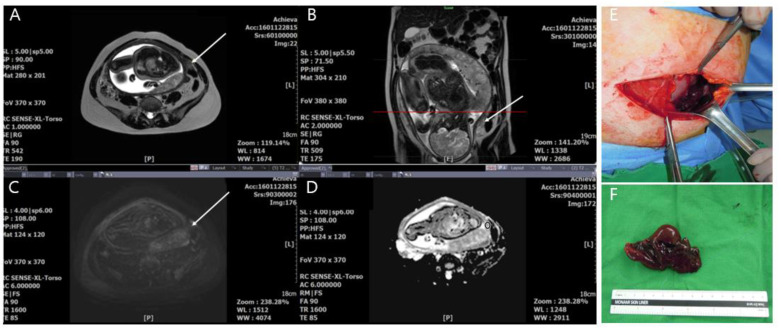
A 34-year-old pregnant woman (35.2 weeks) with acute salpinx torsion with hemorrhagic infarction in the salpingectomy group. (**A**,**B**) Transaxial and sagittal T2-weighted fast spin-echo MR images show slightly swollen ovary (not seen in this figure) and hyperintense fallopian tube or fimbriae (arrow). (**C**) Transaxial diffusion-weighted MR image shows heterogenous slightly hyperintense swollen fallopian tube (arrow). (**D**) Apparent diffuse coefficient (ADC) map shows diffusion restriction of fallopian tube or fimbriae (arrow). ADC value in a circle indicating the region of interest (ROI) placed in fallopian tube or fimbriae was 0.96 ± 0.13 × 10^−3^ mm^2^/s. (**E**,**F**) Intraoperative photograph reveals a necrotic fallopian tube or fimbriae (arrowhead) under laparotomy. (Color version of the figure).

**Figure 4 jcm-09-02209-f004:**
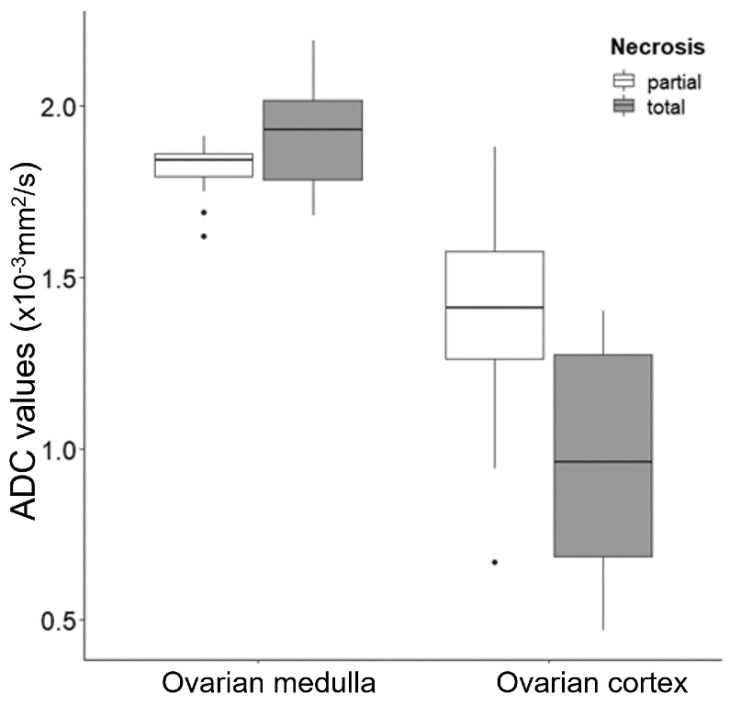
Box-and-whisker plot of ADC values for partial and total necrosis between ovarian medulla and cortex, respectively. The difference in ovarian cortex groups was significant. (*P* = 0.022). ADC, apparent diffusion coefficient.

**Figure 5 jcm-09-02209-f005:**
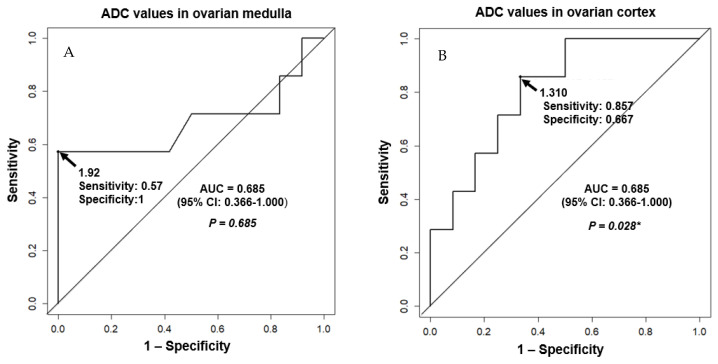
Receiver operating characteristic curves of ACD for predicting the severity of hemorrhagic infarction in the torsed maternal ovary during pregnancy. Receiver operating characteristic curves for necrosis based on ADC values between ovarian medulla and cortex for predicting ovarian necrosis of maternal adnexal torsion during pregnancy are shown. (**A**) Optimal cut-off ADC value for predictable total necrosis in torsed ovarian medulla was ≤1.92 × 10^−3^ mm^2^/s (area under the curve (AUC) = 0.685; 95% CI 0.366–1.0; *P* = 0.685). Sensitivity of 57.1% (4/7), The specificity of 100.0% (12/12), PPV of 100.0% (4/4), NPV of 80.0% (12/15), false positive of 0.0% (0/12), false negative of 42.9% (3/7) was obtained, respectively. (**B**) Optimal cut-off ADC value for predictable total necrosis in torsed ovarian cortex was ≤ 1.31 × 10^−3^ mm^2^/s (area under the curve (AUC) = 0.81; 95% CI 0.611–1.0; *P* = 0.028). The sensitivity of 85.7% (6/7), specificity of 66.7% (8/12), PPV of 60.0% (6/10), NPV of 88.9% (8/90), false positive of 33.3% (4/12), false negative of 14.3% (1/7) were obtained, respectively. * represents significant results. ADC, apparent diffusion coefficient; AUC, area under the curve; ROC, Receiver operating characteristic curves; PPV, positive predictive value; NPV, negative predictive value; 95% CI, 95% confidence interval.

**Figure 6 jcm-09-02209-f006:**
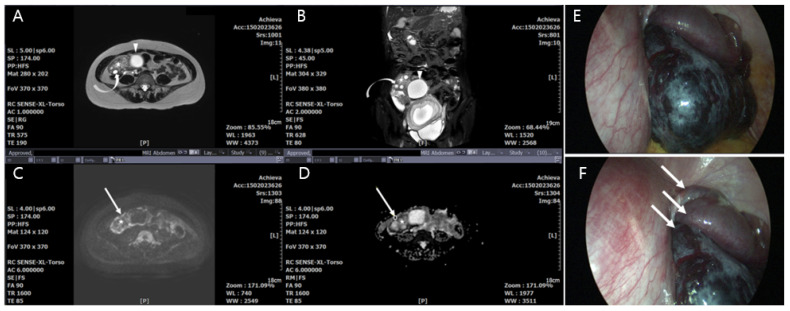
A 31-year-old pregnant woman (11 weeks) with acute adnexal torsion with hemorrhagic infarction in a salpingo-oophorectomy group. (**A**,**B**) Tranaxial and sagittal T2-weighted single-shot turbo spin-echo MR images show hyperintense swollen ovarian medullary stroma (star) and prominent cortical follicles (curved arrow). A 4.5 cm sized unilocular cystic mass (arrowhead) was pathologically diagnosed as a corpus luteal cyst. (**C**) Transaxial diffusion-weighted MR image shows the hyperintense cortex (arrow) and heterogenous hyperintense swollen ovarian medulla (star). (**D**) Apparent diffuse coefficient (ADC) value indicating the region of interest (ROI) was 1.97 ± 0.11 × 10^−3^ mm^2^/s in the ovarian medulla (star), and 0.47 ± 0.05 × 10^−3^ mm^2^ in the ovarian cortex (circle), respectively (**E**) and (**F**) Gross appearance of the ovarian pedicle with 1080° torsion (arrows) under laparoscopy.

**Table 1 jcm-09-02209-t001:** Demographics of the participating pregnant women.

Characteristic	Pregnant Women Diagnosed with Adnexal/Non-Adnexal Torsion				
	Adnexal Torsion Patients Proved by Surgical Confirmation (*n* = 19) a	Non-Adnexal Torsion Patients Proved by Surgical Confirmation (*n* = 9) b	Surgically Unavailable Adnexal/Non-Adnexal Torsion Patients (*n* = 9) c	*P*-Value	Post Hoc
Maternal age, (mean ± SD), yrs	31.68 ± 3.18	30.89 ± 4.31	32.22 ± 3.93	0.843	
Primiparous, *n* (%)					
Negative	7 (36.8)	3 (33.3)	5 (55.6)	0.674	
Positive	12 (63.2)	6 (66.7)	4 (44.4)		
Parity, (mean ± SD)	0.47 ± 0.70	0.33 ± 0.50	0.56 ± 0.53	0.667	
ART					
Natural	12 (63.2)	7 (77.8)	8 (88.9)	0.029	-
IVF + ET	7 (36.8)	0 (0.0)	1 (11.1)		
COH + IUI	0 (0.0)	2 (22.2)	0 (0.0)		
Gestational age, (mean ± SD), wks	15.98 ± 8.55	18.92 ± 10.20	18.40 ± 10.89	0.701	
Trimester *, *n* (%)					
First (<14)	12 (63.2)	4 (44.4)	3 (33.3)	0.620	
Second	4 (21.1)	2 (22.2)	3 (33.3)		
Third	3 (15.8)	3 (33.3)	3 (33.3)		
Trimester, %					
<24 weeks	16 (84.2)	6 (66.7)	5 (55.6)	0.230	
>24.1weeks	3 (15.8)	3 (33.3)	4 (44.4)		
Multigestation					
Singleton	16 (84.2)	8 (88.9)	9 (100.0)	0.790	
Twin pregnancy	3 (15.8)	1 (11.1)	0 (0.0)		
Ovary size, (mean ± SD), cm	6.71 ± 4.03	11.50 ± 5.15	4.48 ± 2.43	0.011	b > c
OHSS					
Negative	15 (78.9)	8 (88.9)	9 (100.0)	0.566	
Positive	4 (21.1)	1 (11.1)	0 (0.0)		
Laterality of tumor					
Rt	12 (63.2)	4 (44.4)	7 (77.8)	0.081	
Lt	7 (36.8)	2 (22.2)	2 (22.2)		
Bilateral	0 (0.0)	3 (33.3)	0 (0.0)		
Mode of Surgery					
Laparotomy	3 (16.7)	5 (71.4)		0.017	
Laparoscopy	15 (83.3)	2 (28.6)			
Mode of Surgery					
Laparotomy	3 (15.8)	5 (55.6)		0.011	
Laparoscopy	15 (78.9)	2 (22.2)			
laparoscopically assisted laparotomy	1 (5.3)	2 (22.2)			
Type of Surgery					
Cystectomy or Detorsion + Cystectomy	12 (63.2)	7 (77.8)	0 (0.0)	0.001	
Salpigectomy	2 (10.5)	1 (11.1)	0 (0.0)		
Oophorectomy	5 (26.3)	1 (11.1)	0 (0.0)		
Conservative treatment	0 (0.0)	0 (0.0)	9 (100.0)		
Body mass index	22.92 ± 4.69	23.40 ± 4.01	27.36±8.60	0.185	
Body mass index, %					
<25.0 kg/m^2^	16 (84.2)	7 (77.8)	5 (55.6)	0.507	
25.0–29.9 kg/m^2^	1 (5.3)	1 (11.1)	2 (22.2)		
30.0–39.9 kg/m^2^	2 (10.5)	1 (11.1)	1 (11.1)		
>40 kg/m^2^	0 (0.0)	0 (0.0)	1 (11.1)		
Degree of necrosis of Torsion					
Partial necrosis	12 (63.2)			-	
Total necrosis	7 (36.8)				
Fetal outcome					
Normal delivery, %	13 (68.4)	5 (55.6)	4 (44.4)	0.226	
Cesarean delivery, %	1 (5.3)	3 (33.3)	1 (11.1)		
Previous cesarean delivery, %	5 (26.3)	1 (11.1)	3 (33.3)		
Spontaneous abortion, %	0 (0.0)	0 (0.0)	1 (11.1)		
Birth weight, (Mean ± SD(g)	3,101.05 ± 550.98	3,180.00 ± 433.76	3,328.75 ± 578.88	0.600	
Weeks at delivery, (mean ± SD), wks	37.93 ± 1.17	38.08 ± 1.36	35.04 ± 9.54	0.798	
Spontaneous abortion	0 (0.0)	0 (0.0)	1 (11.1)	0.087	
Preterm delivery	1 (5.3)	3 (33.3)	1 (11.1)		
Full-term delivery	18 (94.7)	6 (66.7)	7 (77.8)		
Time from pre-operative first MRI performance to surgery (h)	8.42 ± 11.80	437.11 ± 560.87		0.020	a < b
Time to discharge (days)	4.89 ± 2.11	4.67 ± 2.50	6.44 ± 6.78	0.893	
Tocolytics after surgery					
not use	16 (84.2)	8 (88.9)	6 (66.7)	0.543	
use	3 (15.8)	1 (11.1)	3 (33.3)		
Operation time (mean ± SD), min	52.00 ± 16.24	89.67 ± 54.00		0.009	a < b
Blood loss (mean ± SD), mL	40.79 ± 9.47	124.44 ± 120.53		0.049	a < b
WBC (mean ± SD)	11,081.05 ± 3,336.28	9,794.44 ± 2,530.09	14,313.33 ± 12,325.22	0.560	
Hospital day (mean ± SD), days	4.89 ± 2.11	4.67 ± 2.50	6.44 ± 6.78	0.893	

* First trimester defined as 0–13 6/7 weeks of gestation from last menstrual period; second trimester defined as 14 0/7–27 6/7 weeks of gestation from last menstrual period; third trimester defined as 28 0/7–42 0/7 weeks of gestation from last menstrual period. assisted reproductive technology, ART; IVF-ET, in vitro fertilization in vitro fertilization and embryo transfer; COH, controlled ovarian hyperstimulation: IUI, intrauterine insemination; OHSS, Ovarian hyperstimulation syndrome; WBC, white blood cell; Wks, weeks; h, hours.

**Table 2 jcm-09-02209-t002:** Diagnostic analysis of ultrasonography image and MRI in maternal adnexal torsion during pregnancy.

Test Characteristics	Ultrasonography Image	MRI
	Value (95% CI)	Value (95% CI)
Prevalence	72.7% (49.8–89.3%)	67.9% (47.6–84.1%)
Sensitivity	62.5% (35.4–84.8%)	100.0% (82.4–100.0%)
Specificity	83.3% (35.9–99.6%)	77.8% (40.0–97.2%)
Positive predictive value	90.9% (58.7–99.8%)	90.5% (69.6–98.8%)
Negative predictive value	45.5% (16.7–76.6%)	100.0% (59.0–100.0%)
False positive	16.7% (0.4–64.1%)	22.2% (2.8–60.0%)
False negative	37.5% (15.2–64.6%)	0.0% (0.0–17.6%)
ROC area	0.73 (0.50–0.89)	0.89 (0.71–0.98)

MRI, magnetic resonance imaging; ROC, receiver operating characteristics; 95% CI, 95% confidence interval.

**Table 3 jcm-09-02209-t003:** MR imaging features for predicting necrosis amid maternal adnexal torsion during pregnancy.

MRI Torsion Features	Adnexal Torsion Patients Proved by Surgical Confirmation (*n* = 19)	Partial Necrosis	Total Necrosis	
		Surgical Procedure		*P*-Value Derived from Univariate Analysis
		Cystectomy and Detorsion (*n* = 12)	Salpigno-Oophorectomy (*n* = 7)	
Mean diameter of the cystic mass with the ovary or adnexal mass * (cm)	6.71 ± 4.03	7.50 ± 4.02	5.36 ± 3.96	0.108
Whirlpool sign (a twisted ovarian pedicle or twisted fallopian tube)	19 (100.0)	12 (100.0)	7 (100.0)	NA
Tubal wall thickness	19 (100.0)	12 (100.0)	7 (100.0)	NA
Symmetrical or asymmetrical ovarian cystic wall thickness	19 (100.0)	12 (100.0)	7 (100.0)	NA
Ovarian stromal (medullary) edema	19 (100.0)	12 (100.0)	7 (100.0)	NA
Prominent follicles in in the periphery of the ovarian parenchyma (4 mm or more)	11 (57.9)	6 (50.0)	5 (71.4)	0.633
Periadenexal fat stranding	16 (84.2)	11 (91.7)	5 (71.4)	0.523
Uterine deviation to the twisted side	4 (21.1)	2 (16.7)	2 (28.6)	0.603
Peritoneal fluid	8 (42.1)	5 (41.7)	3 (42.9)	1.00
ADC values in ovarian medulla * (10^−3^ mm^2^/s)	1.85 ± 0.13	1.81 ± 0.09	1.91 ± 0.18	0.209
ADC values in ovarian cortex * (10^−3^ mm^2^/s)	1.22 ± 0.38	1.37 ± 0.32	0.96 ± 0.36	0.022

* Data are given as mean ± SD (range). NA, not available; ADC, apparent diffusion coefficient.

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
