# Peer review of "The Diagnostic Accuracy of Magnetic Resonance Imaging for Maternal Acute Adnexal Torsion during Pregnancy: Single-Institution Clinical Performance Review"

_jcm, 2020, doi:10.3390/jcm9072209_

Round 1
Reviewer 1 Report
The authors aimed to evaluate the accuracy of MRI for the diagnosis of adnexal torsion in pregnancy
The research question is interesting albeit few points are to be discussed
- Conclusion section in the abstract does not relate to the study question about accuracy of MRI.
- The authors conclude the introduction section by mentioning that the current literature is poor with data regarding assessing ultrasound for the diagnosis of of torsion in pregnancy.Recently additional information was added but not mentioned in your reference.
- Percentages are given for SPECIFICITY and NPV. In your results you describe the patients that did not have operation ( pain relief) since some patients were not operated true specificity can not be calculated unless all were operated
- What did you mean by 9 patients in whom in operation "was not clear" if there was adnexal torsion or not?
- Most MRI images are not clear for the reader
- Please clarify what do you mean by Prevalence in Table 2
- It will be interesting to read in the discussion section a reasonable explanation for the limitation of ultrasound in diagnosis of torsion as sign like edema,whirlpool are known to be clearly demonstrated sonographically so were comes the true added value of the MRI?
- Who performed the ultrasound examination -was it an expert? resident? gynecologist?
- After all only 37 patients with only 19 patients with torsion and 9 without limits the main conclusion of advantage of MRI
Reviewer 2 Report
Thank you very much for allowing me to review the manuscript entitled : „The diagnostic accuracy of magnetic resonance imaging for maternal acute adnexal torsion during pregnancy: Single-institution clinical performance review”.
The purpose of the study was to determine the efficacy of diagnostic MRI modality using multiparameter for maternal adnexal torsion during pregnancy.Maternal adnexal torsion in pregnancy is relatively rare, with an incidence below 5%, but incidence could rise to even more than 12% in cases of ovarian hyperstimulation related to associated reproductive technology.
An accurate diagnosis is particularly important for choice of the optimal treatment methods and management in pregnant women , because may be associated with pregnancy complications. May lead to preterm delivery and prematurity. Early diagnosis is very important for optimal time of treatment and may prevent both maternal and fetal / neonatal complications. It is sometimes challenges in perinatal medicine. One of the questions is important: which tools (USG, Doppler examination or CT or MRI ) is the best tool in pregnant women.
Adnexal torsion in pregnancy is a particularly unique disease entity. it seems that the approach to timely intervention for adnexal torsion in pregnancy, tmay be similar to that used when treating nonpregnant women.there seems to be no study that has attempted to evaluate the accuracy of USG or MRI in maternal adnexal torsion during pregnancy.
This study shows that the goal of MRI is to obtain a precise diagnosis that guides the therapeutic decision regarding the most appropriate treatment.
The Authors concluded that maternal adnexal torsion during pregnancy occurred in most corpus luteal cystic ovary cases and some normal-appearing ovary during the 1st and 2nd trimesters of gestation and that ADC signal intensity was confirmed to predict the severity of hemorrhagic infarction in the torsed ovary.
Treatment guidelines for adnexal torsion during pregnancy, ranging from the imaging diagnosis modality to surgical treatment, are still controversial due to the rarity of this condition in pregnancy.
These data suggest that MRI is useful not only for identifying or excluding adnexal torsion during pregnancy but also for providing
a predictive diagnosis to decide further treatment.
In my opinion, the Authors raise a very important topic and relevant issue.
It is a interesting paper very important for our clinical work, which I recommend for publication. This is a well done study . Te manuscript is very interesting and important , original and high scientific value ( writing style - concise , and clear; the discussion is nice , list of references appropriate.
This paper is well written. The text of this manuscript is clear and easy to read.
An additional advantage of the work is photographic documentation that allows the reader to better receive the data presented in the manuscript.
This is a nicely done study and paper focused on an important issue.
This study has several limitations / a few specific concerns:
First, this study is retrospective.
Second limitation of this study is relatively small sample size.
Author Response
I feel honored to submit our manuscript, “The diagnostic accuracy of magnetic resonance imaging for maternal acute adnexal torsion during pregnancy: Single-institution clinical performance review” on behalf of my co-authors.
Thank you again for a good review.
Reviewer 3 Report
- Generally supportive
- As a reviewer had difficulty following the numbers re flow process US first yes / no then those that needed further imaging yes / no then the surgical findings Figure 1 needs to include both US and MRI
- Could not sort out the patients re Table 1 and Figure 1
- The imaging comments are fine but this does not really allow the clinician to use a clinical process; Are you saying if torsion is suspected go directly to MRI ; it just is not clear to me.
- Discussion is OK but the limitations need to be expanded as the number of final cases and surgically unavailable are very hard to follow.
Round 2
Reviewer 3 Report
Comments on Revision 1: All the concerns identified have been corrected